# Spatial release from masking in crocodilians

Julie Thévenet [1,2,5✉], Léo Papet [1,2,5✉], Zilca Campos [3], Michael Greenfield [1,4], Nicolas Boyer[1], Nicolas Grimault [2,6] & Nicolas Mathevon [1,6]

Ambient noise is a major constraint on acoustic communication in both animals and humans. One mechanism to overcome this problem is *Spatial Release from Masking* (SRM), the ability to distinguish a target sound signal from masking noise when both sources are spatially separated. SRM is well described in humans but has been poorly explored in animals. Although laboratory tests with trained individuals have suggested that SRM may be a widespread ability in vertebrates, it may play a limited role in natural environments. Here we combine field experiments with investigations in captivity to test whether crocodilians experience SRM. We show that 2 species of crocodilians are able to use SRM in their natural habitat and that it quickly becomes effective for small angles between the target signal source and the noise source, becoming maximal when the angle exceeds 15°. Crocodiles can therefore take advantage of SRM to improve sound scene analysis and the detection of biologically relevant signals.

[1] Equipe de Neuro-Ethologie Sensorielle ENES / CRNL, CNRS, Inserm, University of Saint-Etienne, Saint-Etienne, France. [2] Equipe Cognition Auditive et Psychoacoustique / CRNL, CNRS, Inserm, University Lyon 1, Bron, France. [3] Wildlife Laboratory, Brazilian Agricultural Research Corporation EMBRAPA, Corumbá, Brazil. [4] Department of Ecology and Evolutionary Biology, University of Kansas, Lawrence, KS 66045, USA. [5] These authors contributed equally: Julie Thévenet, Léo Papet. [6] These authors jointly supervised this work: Nicolas Grimault, Nicolas Mathevon. ✉email: julie.thevenet@inserm.fr; leo.papet@univ-st-etienne.fr

Animals that use acoustic signals to communicate often develop strategies for optimizing information transfer in noisy soundscapes[1–6]. Emitters may increase Signal-to-Noise Ratio (SNR) by raising signal intensity (Lombard effect[7]), by shifting signal frequency to avoid overlap with the noise frequency bandwidth (e.g. in great tits *Parus major* and zebra finches *Taeniopygia guttata*[8,9]), by using signal redundancy[10–12], or by choosing emission posts and behavioral postures that optimize signal transmission (e.g. songposts[13–15]). At the other end of the communication chain, receivers may choose strategic posts and behaviors that improve signal reception and facilitate auditory computation in noisy environments (e.g. hearing posts in songbirds[16,17]). When listening in noise, spatial cues such as Interaural Time Differences (ITD) and Interaural Level Differences (ILD) play an important role in improving signal detection, source localization, and information decoding[18–21]. Spatial Release from Masking (SRM) refers to the process where the auditory system of listeners uses these directionally dependent cues to segregate the signal of interest (target) from competing sounds (maskers[22,23]). According to SRM, signal reception is better when the signal source is spatially separated from the noise source than when both signal and noise sources are co-located in the environment[23,24].

SRM has primarily been investigated in humans. The seminal study by Saberi et al.[24] demonstrated that SRM is efficient in both the horizontal and vertical planes in our species[25]. SRM has also been found in a few other mammal species: ferrets *Mustela putorius*[26], cats *Felis catus*[27], big brown bats *Eptesicus fuscus*[28], harbor seal *Phoca vitulina*, and sea lion *Zalophus californianus*[29]. In birds, SRM enhances the detection of pure tones masked by a broadband noise in budgerigars *Melopsittacus undulatus*[30], and the detection of bird songs in a song chorus in both zebra finches *Taeniopygia guttata* and budgerigars[31]. SRM has been investigated in amphibians (northern leopard frogs *Rana pipiens pipiens*[32], Cope's gray treefrog *Hyla chrysoscelis*[33–37]), showing better detection and discrimination of conspecific calls masked by noise when the two sources are spatially separated[22,38]. Finally, SRM has also been found in two crickets (*Paroecanthus podagrosus* and *Diatrypa sp.*), where it improves the detection of natural conspecific song against the ambient noise of the rainforest[39]. Notably, the fly *Ormia ochracea* is the only known animal species which seems not able to benefit from SRM[40]. In addition to sound communication in air, SRM has also been found in underwater communication with bottlenose dolphins *Tursiops truncatus*[41].

Although SRM could appear as a widespread ability to increase the detection of sound signals against masking noise, it has yet been investigated in a limited diversity of experimental approaches and situations. All previous studies investigating SRM in animals have been performed in very controlled conditions in the laboratory or captive environments[26–31,34,38,41,42]. There has been no field investigation with animals freely behaving in their natural habitat. This is a serious limitation: it cannot be ruled out that SRM is a laboratory artifact with a limited role in the field. Indeed, in the field, animals are exposed to a wider and more realistic range of situations, e.g. in terms of head position relative to the sound source and noise. Testing their SRM abilities in field condition would certainly provide a more realistic picture. Moreover, all studies performed in vertebrates (except one with treefrog[38]) have been based on conditioning experiments where animals were trained to locate sound sources (Go/No-Go experiments[26–31,34,42]). While Go/No-Go experiments may limit the variability of the tested subjects' motivation, an intensive training combined with laboratory conditions is likely to change the ability of subjects to perform SRM compared to natural field conditions. Strikingly there has been no study on SRM combining different experimental approaches, in both controlled and natural settings. In spite of its tremendous utility for sound scene analysis in the daily life of animals, SRM thus remains a poorly investigated phenomenon.

In the present study, we investigated SRM in crocodilians. These animals may indeed be ideal subjects for studying SRM in various conditions for the following reasons. First, they are relatively immobile, which allows us to conduct these acoustic experiments in the field with a precision and a control of initial conditions usually restricted to laboratory experiments. Second, they do actively use acoustic communication during their social interactions[43,44], where the detection of signals could be critical. Mature embryos vocalize to synchronize hatching and promote maternal care[45]. Juveniles emit contact calls ensuring group cohesion, and distress calls inducing maternal protection[46,47]. Adult males of most species attract females and repel competitors by producing a repertoire of vocalizations (bellows, grunts) as well as low frequency sounds through the vibration of their whole body[48], while females emit grunts to attract their young[43]. Third, crocodilians spend most of their active life cruising at the interface of air and water. In this amphibious environment, they can be exposed to various sources of noise, either biotic (e.g. chorusing frogs) or abiotic (e.g. waterfall noise, anthropogenic noise such as boats). This noise may mask crocodilians' vocalizations and may thus impair their acoustic communication. The receiving individual must discriminate the signal of interest against non-relevant masking sounds, and SRM could represent a valuable ability. Moreover, the head morphology of crocodilians enables them to acquire reliable localization cues from sound sources propagating in the air even when only a small part of their head is above the air-water interface[49,50]. In a previous study, we found that crocodiles may use both Interaural Level Differences cues and Interaural Time Differences cues to accurately locate the spatial direction of a sound source[51–53]. However, the radically different acoustic impedances of air and water prevent most of the acoustical energy from entering the water and thus removes part of the acoustical difference between right and left ears[49,50].

Here we demonstrate that crocodilians cruising in water use SRM to detect target sounds against a noisy background both propagating in the air. We used three different experimental paradigms to explore this ability. First, we examined whether adult crocodilians (*Caiman yacare*) use SRM in natural conditions by performing field experiments in the Pantanal, Brazil. We challenged naive caiman mothers while they were caring for their young by mimicking a situation where an isolated nestling was emitting distress calls[54]. We then tested whether these SRM abilities are already present in young crocodilians with experiments in a zoo, where we assessed the response of naive young Nile crocodiles *Crocodylus niloticus* to the playback of contact calls[46]. Jacare caimans and Nile crocodiles are two representatives of two of the three extant groups of crocodilians that differentiated during the Cretaceous: the Alligatoroidea and Crocodyloidea respectively (the third group being the Gavialoidea). Finally, we tested whether SRM functions with non-biological signals in the laboratory by training juvenile Nile crocodiles to identify a synthesized sound from a masking noise using Go/No-Go experiments. In these three experimental situations, we evaluated the ability of the tested individuals to detect the source of the target signals as a function of the location of the background noise source.

## Results

**Spatial release from Masking by adult crocodilians.** This first experiment was conducted on wild adult female yacare caimans

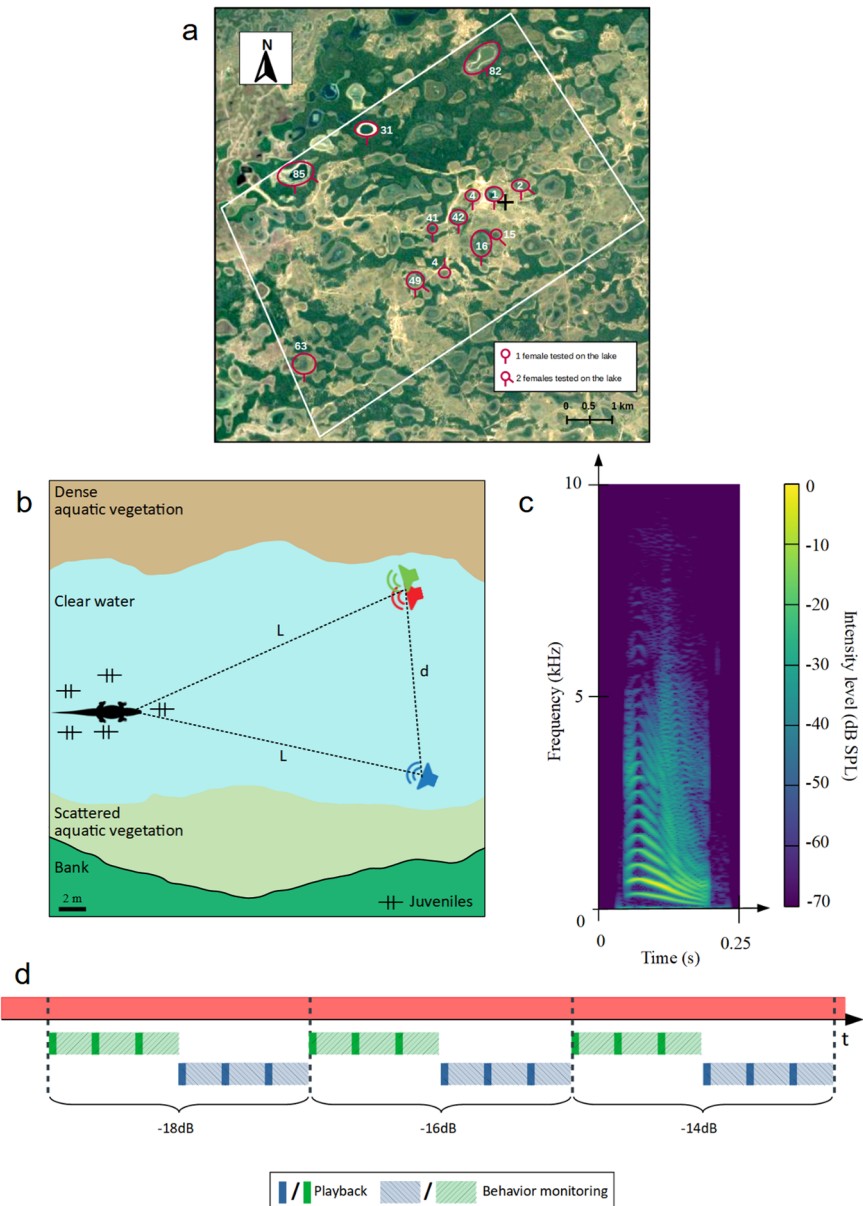

**Fig. 1 Field experiments on female Jacare caimans (Experiment 1). a** Cartography of the field work area (white rectangle = border of the Nhumirim reserve; black cross = field station; 18°59'16.1"S 56°37'08.8"W). We conducted the experiments in the lakes surrounded by red circles. The number of red hyphens indicates the number of females tested on the same lake (1 or 2 individuals). **b** Schematic representation of the experimental design. The distance between the loudspeakers (d ∈ [6.5, 19]m) was always lower than the distance between the female and the loudspeakers (L ∈ [12, 50]m). **c** Spectrogram of a distress call from a young Jacare caiman. **d** Timeline of an experiment. The masking noise is emitted continuously (red solid line). The target signals (with different SNR) are emitted either by the co-located loudspeaker (green solid line) or by one of the separated loudspeakers (blue solid line). The light dashed lines represent the behavior recording following the target emission.

*Caiman yacare* (Pantanal, Brazil, Fig. 1). For each female (N = 16), we played back a broadband noise ("noise source", emitted at 83 dB SPL, unweighted) simultaneously with a series of distress calls recorded from young juveniles ("target signal"; relative intensity to the noise in the range [−20, 0] dB; calls recorded from 3-week old individuals, unfamiliar to the tested females, see Fig. 1c for the spectrogram of one call). In this experiment, distress calls were selected to optimize a behavioral response from the female toward the loudspeaker[43]. As illustrated on Fig. 1b, the two loudspeakers emitting the "noise source" and the "target signal" were either side-by-side ("co-located condition") or spaced apart ("separated condition"; mean separation angle between the female, the noise source and the distress calls

source = 18°, min–max = 4–44°, Raw data in Supplementary Fig. 1a).

Each female was successively challenged with several co-located and separated target signals with various Signal-to-Noise Ratios (SNR) and separation angles (16 females tested, mean number of trial per female = 6.5 ± 5; Fig. 1d, see Supplementary Fig. 1a for raw data and Supplementary Table 1 for details on the signals played back to each female; see Supplementary Movie 1 for a video showing an experimental trial).

At the beginning of each playback, the loudspeakers were at approximately 20 meters from the tested female. We rated the female's response to playback according to a 0–4 level behavioral scale (score for no reaction = 0; head or body movement not in

the direction of the target loudspeaker = 1; head movement towards the target loudspeaker = 2; displacement on a distance less than 1 body length towards the target loudspeaker = 3; displacement on a distance more than 1 body length towards the target loudspeaker = 4). We compared the behavioral reactions between experimental conditions using a Bayesian approach (see Methods for details). In summary, the probability of behavioral scores was fitted according to two different models: one with the SNR and the position of target source (i.e. either co-located or separated) as fixed factors, and another one with only the data obtained in the separated condition, with the SNR and the initial angle of separation between the target and the masker as fixed factors.

The playbacks revealed that the female's response depended on the SNR between the target and the masker, with higher SNRs inducing higher behavioral scores (Bayesian ordinal model: $\beta_{SNR} = 0.32$, 95% CI = [0.21, 0.45], probit scale; Supplementary Table 2, Supplementary Fig. 1a for raw data and 2a). They also provided strong evidence of an effect of the separation of the target source from the masking noise source, with higher behavioral scores being more likely in the separated condition compared to the co-located condition ($\beta_{separation} = 1.15$, 95% CI = [0.63, 1.69]; Supplementary Fig. 2b). By fitting the probabilities of the behavioral scores in function respectively of the SNR and of the co-located and separated conditions, we confirmed that the females' motivation to move towards the target loudspeaker depended highly on the SNR, with lower SNR levels eliciting a female reaction in the separated condition (Fig. 2a, b). Accordingly, the separated condition decreased both the SNR threshold from which the females began to respond and the SNR threshold eliciting a full response (Supplementary Table 3).

The signal detection threshold (SNR value corresponding to a 50% probability of a behavioral score equal or higher than 1) was −14.6 dB in the co-located condition while it dropped to −18.2 dB in the separated condition. The full response threshold (SNR value corresponding to a 50% probability of a behavioral score of 4) was equal to −9.1 dB in the co-located condition and −12.7 dB in the separated condition. Both thresholds (i.e. signal detection threshold and full response threshold) lead to a SRM effect equal to 3.6 dB. Interestingly, we found neither an effect of the SNR ($\beta_{SNR} = -0.06$, 95% CI = [−0.41, 0.27], skew normal distribution; Supplementary Fig. 5a) nor of the relative positions of the target and noise sources relative positions ($\beta_{separation} = -0.88$, 95% CI = [−3.93, 2.44]; Supplementary Fig. 5b) on the females' reaction time once the stimulus is detected (Supplementary Table 4). Thus, while SRM helps the animal to detect a signal in a noisy environment, it does not seem to influence the delay between the detection and the behavioral reaction.

We then tested for an effect of the angle of separation between the target and the noise sources on the females' responses by focusing only on the separated condition (target loudspeaker separated from the masker; min angle = 4°, max angle = 44°). Figure 2c shows the fitted probabilities of each 0–4 behavioral score as a function of the angle of separation, while controlling for the SNR. The results support the hypothesis that the larger the angle, the stronger the female's response ($\beta_{Angle} = 0.10$, 95% CI = [−0.01, 0.23] on the probit scale; 95.9% confidence that higher angles of separation between the target and the masker elicited higher behavioral scores; Supplementary Table 5 and Supplementary Fig. 2c).

**Spatial Release from Masking by juvenile crocodilians**. This second experiment was performed on young naive Nile crocodiles *Crocodylus niloticus* in captivity (3-months old juveniles, N = 8).

We tested their ability to detect a target signal against noise by playing back series of "contact" calls in a noisy environment[46] (see Fig. 3b for the spectrogram of one call). For each experiment, a crocodile was placed in a large outdoor pool (diameter 8 meters) where a loudspeaker placed on the edge of the pool was continuously emitting a broadband noise.

Several hours later during the night, we played back series of target signals from other loudspeakers placed at different locations around the pool (one "co-located" loudspeaker side-by-side to the noise loudspeaker and two "separated" loudspeakers, Fig. 3a; when non null, the angle between the separated loudspeakers and the noise loudspeaker varied between 44° and 156°, see raw data in Supplementary Fig. 1b). Each subject was challenged several times with an interval of at least 10 min between trials (Fig. 3c; 7–11 trials per subject; total of 30 "co-located" and 41 "separated" trials; see Supplementary Movie 2 for an example of a playback experiment; see Supplementary Table 6 for details of the signals played back to each juvenile). For each trial, we assessed the crocodile's ability to detect the target signal against the background noise by rating its behavior according to a binary scale: no orientation towards the loudspeaker emitting the target calls = score 0; orientation towards the loudspeaker = score 1. For the purpose of analysis we further modeled this scoring using a Bayesian logistic regression (Bernoulli distribution).

The playback tests showed that the ability of the juvenile crocodiles to detect the target signal against the background noise depended both on the signal-to-noise ratio, with higher SNRs inducing higher probabilities of detection ($\beta_{SNR} = 0.23$, 95% CI = [0.10, 0.39], logit scale; Supplementary Table 7 and Supplementary Fig. 1b for raw data and 3a), and on the source position, with a higher detection probability when the noise and the target loudspeakers were spatially separated ($\beta_{Separation} = 1.57$, 95% CI = [0.40, 2.90]; Supplementary Fig. 3b). These results are in line with those obtained in the field experiments reported in the previous section of the article.

By modeling the signal detection probability in function respectively of the SNR and of the co-located and separated conditions (Fig. 4), we found that the signal detection threshold (SNR value corresponding to a 50% probability of signal detection; for comparison purpose, this would correspond to a score equal or above 3 in the first experiment) was −18.1 dB in the co-located condition while it decreased to −24.9 dB in the separated condition (i.e. SRM amount equal to 6.8 dB). In accordance with this result, a separated target had a 65.4% probability of being detected for an SNR of −22.1 dB (median value) while this probability was only 28.6% for a co-located target (95% CI = [8.9, 59.8]; Supplementary Table 8).

As for the field experiments, we observed no influence of the SNR or of the position of the target loudspeaker on the reaction time ($\beta_{SNR} = -1.31$, 95% CI = [−3.42, 0.78]; $\beta_{Separation} = -11.44$, 95% CI = [−32.28, 8.74]; Supplementary Table 4 and Supplementary Fig. 5c, d). We further tested whether increasing the angle between the noise and the target loudspeaker from 44° (minimum angle in the separated condition) to 156° (maximal angle) could improve the crocodiles' ability to detect the target signal and found no effect ($\beta_{Angle} = -0.01$, 95% CI = [−0.04, 0.03] on the logit scale; Supplementary Table 9 and Supplementary Fig. 3c).

**Spatial Release from Masking by crocodilians to detect a non-biological signal**. This third experiment was performed in laboratory conditions with two juvenile Nile crocodiles (3 years-old). Prior to the experimental procedure, both crocodiles were trained with a Go/No-Go procedure to swim towards a target loudspeaker emitting a synthesized harmonic complex tone

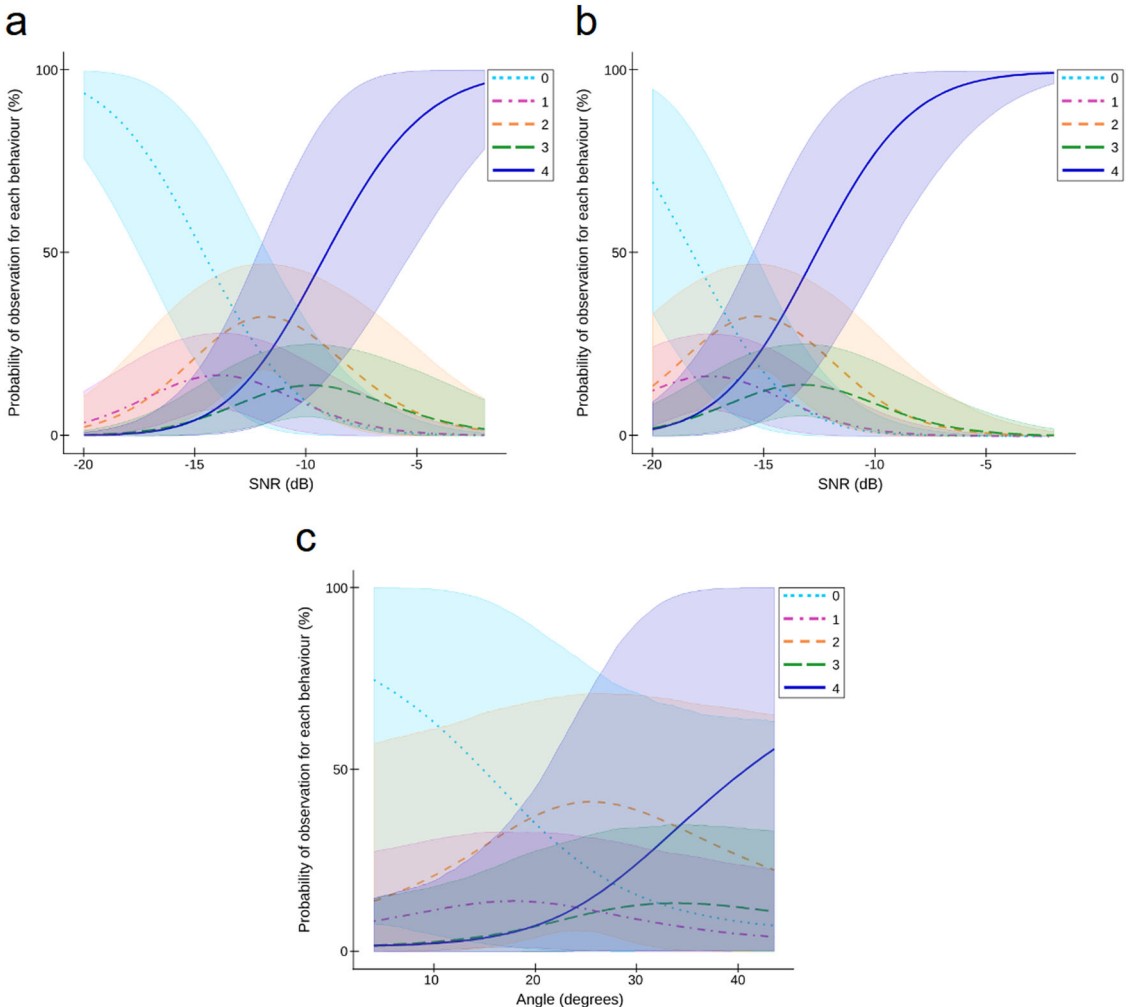

**Fig. 2 Effect of the Signal-to-Noise ratio (SNR) and of the loudspeakers' spacing on the behavioral reaction of female caimans to the playback of young distress calls (Experiment 1, Pantanal, field conditions with wild animals). a** Effect of the SNR on the females' response to sound stimuli when the target and the noise loudspeaker are at the same location ("co-located" condition). The probability of eliciting a higher behavioral response increases with SNR (fitted probabilities of behavioral scores: mean of posterior distribution and 95% credible intervals). **b** Effect of the Signal-to-Noise ratio (SNR) on the females' response when the target and the noise loudspeaker are spaced by a minimum angle of 4° ("separated" condition, mean angle between loudspeakers = 18°, min-max = 4-44°). The females' behavioral reactions are elicited by stimuli with lower SNR compared to the "co-located" condition, supporting the hypothesis that the tested females perform Spatial Release from Masking. **c** Effect of the speaker spacing on the females' response in the "separated" condition. The probability of the females approaching the loudspeaker increases as the separation angle between the target and the noise loudspeakers increases.

(buzz, Fig. 5b; see Methods). The crocodiles' ability to detect the target loudspeaker against a background noise was then tested during several experimental sessions (65 and 55 trials with crocodile 1 and 2, respectively; see Supplementary Table 10 for details).

For each session, one of the crocodiles was placed in an experimental pool in a sound-proofed chamber (Fig. 5a). One loudspeaker was continuously emitting white noise. The target signal (sequences of three synthetic signals identical to the ones used during training) was emitted either by the noise loudspeaker (the noise and the target signals were mixed) or by one of two other loudspeakers placed at other locations on the edge of the pool (Fig. 5a, c). The tests were done in complete darkness. For each trial, we assessed the crocodile's ability to detect the target signal against the background noise by rating its behavior according to a binary scale: no orientation towards the loudspeaker emitting the target calls = score 0; orientation towards the loudspeaker = score 1. For analysis purpose, we further modeled this scoring using a Bayesian logistic regression.

The playback tests showed that the SNR of the target stimuli had a strong effect on the crocodiles' ability to detect the signal against the background noise, with higher SNRs inducing higher probabilities of detection ($\beta_{SNR} = 0.20$, 95% CI = [0.10, 0.32], logit scale; Supplementary Table 11 and Supplementary Fig. 1c for raw data and 4a, see Supplementary Movie 3 for an example of an experimental trial).

Although the effect of the target loudspeaker location (co-located versus separated) appeared weaker than in the two previous experiments, there was a 90.7% probability that separated targets were better detected than co-located ones ($\beta_{Separation} = 0.58$, 95% CI = [−0.28, 1.47]; Supplementary Fig. 4b). By modeling the signal detection probability according to, respectively, the SNR and of the co-located and separated conditions (Fig. 6), we found that the signal detection threshold (SNR value corresponding to a 50% probability of signal detection) was −21.8 dB in the co-located condition while it decreased to −24.6 dB in the separated condition (i.e. spatial release from masking equal to 2.8 dB). A separated target had a

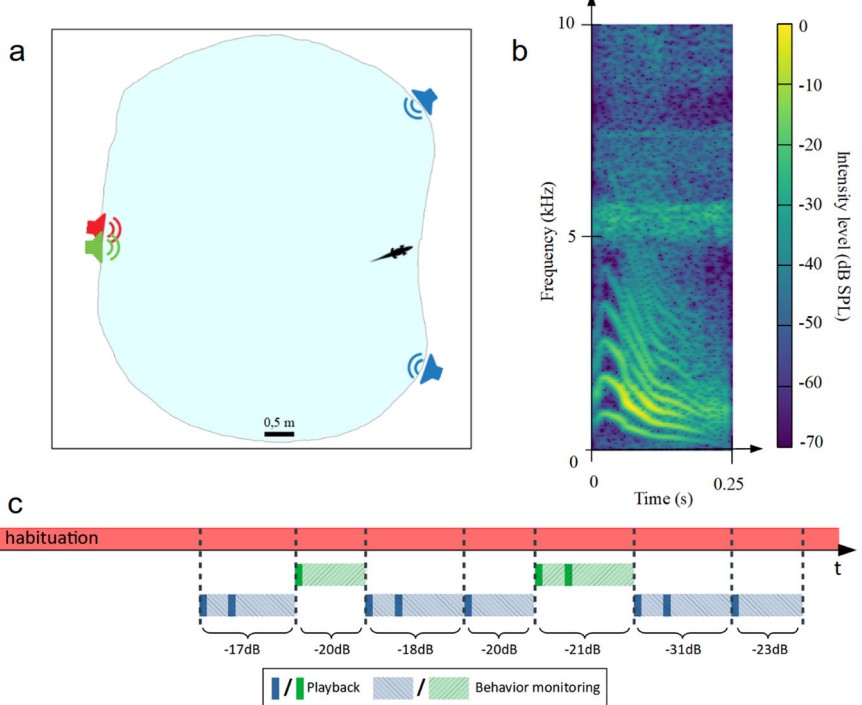

**Fig. 3 Experiments on young Nile crocodiles in captivity (Experiment 2). a** Schematic representation of the experimental design. A noise was continuously emitted by the "noise" loudspeaker (red). The stimuli were emitted either by the "co-located" loudspeaker (green) or one of the "separated" loudspeakers (blue). **b** Spectrogram of a contact call from a young Nile crocodile. **c** Timeline of an experiment. The masking noise is emitted continuously (red solid line). The target signals (with different SNR) are emitted either by the co-located loudspeaker (green solid line) or by one of the separated loudspeakers (blue solid line). The same signal could be played again by the same loudspeaker if the crocodile had not moved 90 seconds after the end of the first emission. The light dashed lines represent the behavior recording following the target emission.

63% probability of being detected for an SNR of −22 dB (median value) while this probability was 49% for a co-located target (95% CI = [−5.70, 32.74] ; Supplementary Table 12).

As in the two other experiments, we found no effect of the SNR or of the angle between the two loudspeakers on the crocodile's reaction time ($\beta_{SNR} = 0.05$, 95% CI = [-0.43, 0.58]; $\beta_{Separation} = 2.06$, 95% CI = [−2.13, 6.72]; Table 5 and Supplementary Fig. 5e, f). We further tested whether increasing the angle between the noise and the target loudspeaker from 16° (minimum angle in the separated condition) to 178° (maximal angle) could improve the crocodiles' ability to detect the target signal and found no effect ($\beta_{Angle} = 0.00$, 95% CI = [−0.01, 0.01]; Supplementary Table 13 and Supplementary Fig. 4c, see Supplementary Fig. 1c for the distribution of angles).

## Discussion

In this study, we tested whether crocodilians use Spatial Release from Masking to detect a sound target against a continuous background noise. We performed experiments in three different contexts: in the field with wild adult animals, in a naturalistic setup in captivity with naive juvenile subjects, and in a laboratory Go/No-Go experiment with trained juvenile subjects. The combined results of these three experimental approaches confirm that crocodilians detect sound signals better when the target source is spatially separated from the masking noise source, suggesting that these animals use SRM in their daily lives.

Conducting experiments with crocodiles can be challenging. In the field and in the zoo, they habituate to played back signals very quickly, which limits the number of trials performed with a given individual. In the field, to ensure as much as possible that each female could be tested in both co-located and separated conditions for several SNR, we choose to present successively the

signals starting from the lowest SNR until it elicits a response from the animal. To avoid a potential cumulative effect due to this protocol, we took several precautions: (1) we were very careful to note the smallest observable behavioural response suggesting a possible detection of the signal, (2) we repeated successively the same signal 3 times to allow the female the opportunity to respond when they hesitated, and (3) we leave a significant temporal delay between the stimuli. In the zoo as in Go/No-Go experiments in the laboratory, we optimized the number of trials by placing several speakers around the ponds in order to change the origin of the sound. Go/No-Go experiments in the laboratory required extensive training of the animals. This time-consuming training, combined with the logistical constraints inherent in these animals when kept in captivity, also limits the number of subjects that can be included in the experiments. In addition, the ectothermy of these animals imposes a long delay between experimental sessions for the animal to regain hunger and be sufficiently motivated to perform the task. These constraints explain why there are some gaps in our data, both in the range of SNRs tested and in the range of separation angles (Supplementary Fig. 1). Such limitations impact the statistical power of the analyses and call for caution in interpreting results. Nonetheless, our data highlight that the spatial separation between the target and noise sources has a major influence on the detectability of the target source. While an increase in SNR improves the signal detection ability of the tested individuals in both types of experimental conditions ("co-located" and "separated"), detection thresholds are always lower when the target and noise sources are spatially separated.

Because the three sets of experiments (field, zoo, and Go/No-Go) differ in terms of speakers' position, distance between speakers and tested individuals, and, most importantly, in terms

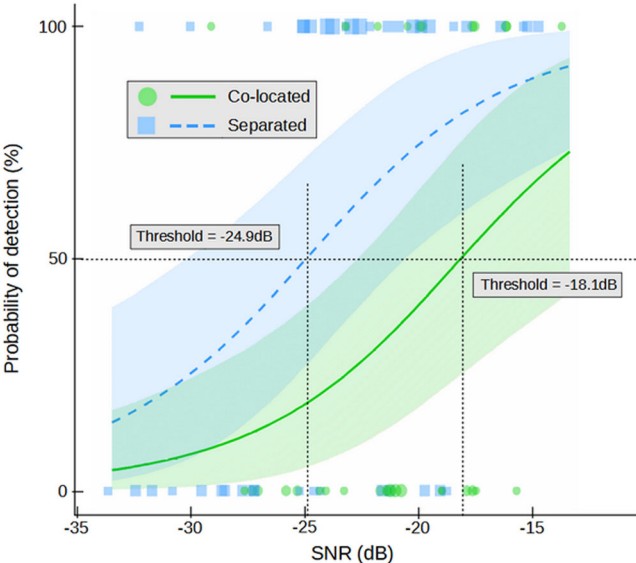

**Fig. 4 Effect of the Signal-to-Noise ratio (SNR) and of the loudspeakers spacing on the behavioral reaction of young Nile crocodiles to the playback of contact calls.** (Experiment 2, Crocoparc Zoo, freely moving animals in a large basin; curves = fitted probabilities of behavioral scores: mean of posterior distribution and 95% credible intervals; green dots represent individual trials in co-located condition, blue squares are individual trials in separated condition). The probability of signal detection increases with SNR in both conditions, i.e. when the target and the noise loudspeaker are close together ("co-located" condition) or spaced apart ("separated" condition). The crocodiles' behavioral reactions are elicited by stimuli with lower SNR in the "separated" condition, supporting the hypothesis that tested young Nile crocodiles perform Spatial Release from Masking. The difference between both detection thresholds (amount of spatial release) is 6.8 dB.

of biological context and crocodilian species, the absolute values of detection and response thresholds cannot be accurately compared. The SRM size effect, however, remains close in magnitude (3.6 dB, 6.8 dB, and 2.8 dB, in the field, in captivity, and in Go/No-Go experiments, respectively), and can be considered representative of crocodilian SRM capabilities.

To the best of our knowledge, this study is the first to investigate SRM in wild animals freely behaving in their natural habitat, and to combine this approach with investigations in captivity and in the laboratory. Both in the field and in the zoo, we did not train the animals to respond. Consequently, the behavioral reaction of the tested subjects to the stimuli was likely modulated by several factors influencing their internal motivation. Therefore, we may have underestimated the ability of animals to detect the target signal, and the amplitude of SRM may be greater than reported. In the field for instance, female Jacare caimans remained close to their own young when challenged with the target signals. Moving toward the target loudspeaker meant that the females had to abandon their young. This trade-off between motivation to stay and motivation to go may have decreased the females' reaction level. It probably explains some of the variation between individual responses. To understand this variation, it would have been interesting to know the number of nestlings present with each tested female, and to monitor the vocal activity of the young as both may have influenced the mother's decision. In the zoo context, juvenile Nile crocodiles who found themselves isolated for the duration of the experiment, may have faced another type of behavioral trade-off between swimming to a speaker mimicking a sibling and remaining still to limit predation risk, as young crocodiles are heavily predated in the wild. Conversely, in the Go/No-Go laboratory experiment, the tested subjects have been trained to move toward the loudspeaker by getting a food reward. Thus, it is likely that the subjects' motivation to respond to the target stimuli was high, and at least, fairly consistent over the course of the experimental trials. Nonetheless, we still observed variability in the animals' response, potentially related to personality differences and also probably to their bradymetabolism differences punctually affecting their motivation to perform the experiment.

Contrary to our expectations, the SRM values obtained with the Go/No-Go procedure were lower than in the two other contexts. One possible explanation lies in the acoustic environment in which the experimental trials were performed. The test booth was quiet (background level = 40 dB SPL), but not perfectly anechoic (reverberation time = 0.44s, volume of the booth = 9.11m³). The tested crocodiles may have perceived some early acoustic reflections in addition to the direct sound waves which may have decreased the ability to detect the target signal.

A second possible explanation for this lower SRM value could come from the nature of the target signal used in this experiment. In humans, the SRM, also related to the cocktail party effect, involves energetic aspects (i.e. energetic masking) as well as cognitive aspects (i.e. informational masking[55]). In the first two experiments, the target signals were biologically relevant to the crocodile, unlike in the last experiment (synthetic buzz). This may have modulated the amount of informational masking across experiments and contributed to the weaker SRM effect in the Go/No-Go experiment.

In humans, the mechanisms underlying SRM have been extensively explored and reviewed[23,25,56]. First, when the target and masker are spatially separated, half of SRM effect comes from the "better ear effect", where the SNR is more favorable in one ear (due to to noise attenuation by the head shadow) than in the other. This effect is purely monaural. Second, the ability of the auditory system to utilize binaural aspects of the signal, including time (ITD) and level (ILD) differences between the ears is also known to contribute to SRM. Third, "binaural summation" (i.e. the fact that a signal presented to the front will activate both ears and then make that sound easier to hear due to the summation of the signals at both ears) provides an additional contribution to SRM. In our study, a reliable SRM effect was reported in all experiments, and all three mechanisms described in human could also have contributed to the observed SRM effect. In particular, it is now well known that crocodile ears are acoustically coupled by air-filled cranial sinuses[21,49], which greatly increases directional cues such as ITDs[57]. At the encoding level, alligators have been shown to form ITD maps in the brainstem nucleus laminaris similarly to birds, again suggesting a convergence among modern archosaurs[53]. The crocodilian binaural system may therefore be as well-developed as that of birds and thus could be effective in detecting spatially separate signals[52]. However, the size of each effect might have been overestimated or underestimated because of the fluctuating position of the crocodile head during stimulation. In fact, under some conditions, the masker and separated target could be played on the same side of the crocodile's head, altering the magnitude of the better-ear-effect and/or the magnitude of the summation effect. Therefore, tested in freely-moving animals, the potential SRM effect may not have been maximized in all trials. In conclusion, on the one hand, the relative contributions of the monaural better-ear-effect, binaural cues and binaural summation effect for SRM remain unknown for non-human animals, including crocodiles, and would require further study. On the other hand, our study supports a global SRM effect in the field, regardless of head position relative to the source and target positions.

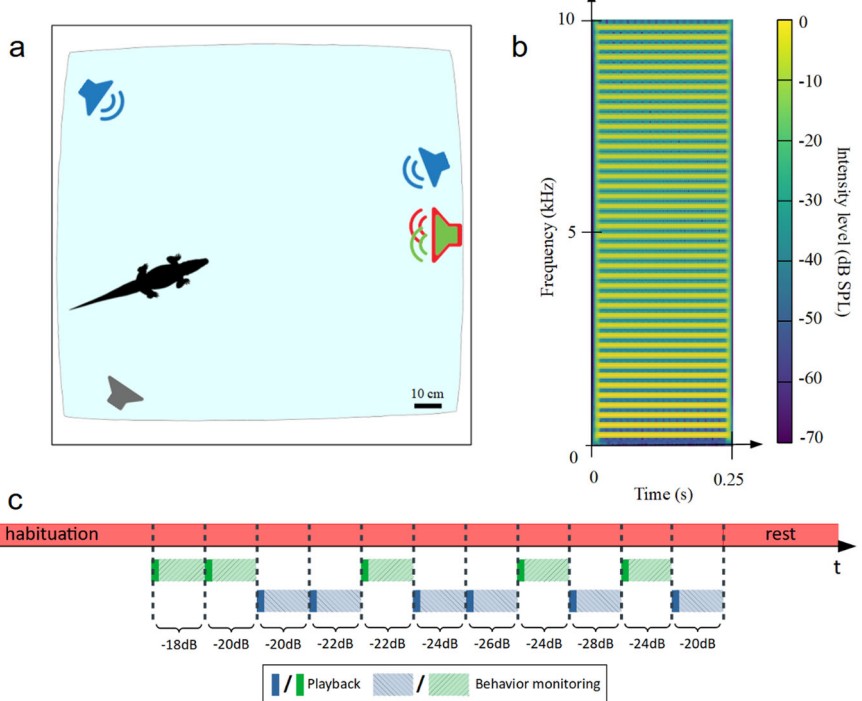

**Fig. 5 Go/No-Go experiments on juvenile Nile crocodiles in captivity (Experiment 3). a** Schematic representation of the experimental design. A noise was continuously emitted by the "noise" loudspeaker (red). The stimuli were emitted either by the "co-located" loudspeaker (green) or one of the "separated" loudspeakers (blue). **b** Spectrogram of the synthetic buzz used as the sound stimulus. **c** Timeline of an experiment. The masking noise is emitted continuously (red solid line). The target signals (with different SNR) are emitted either by the co-located loudspeaker (green solid line) or by one of the separated loudspeakers (blue solid line). The same signal could be played again from the same loudspeaker if the crocodile had not moved 45 seconds after the end of the first emission. The light dashed lines represent the behavior recording following the target emission.

Despite these differences between the three experimental conditions, our results highlight the importance of loudspeaker spacing for each of the three. Strikingly, the field experiment shows that the SRM increases significantly when the separation angle between the target and the noise loudspeakers increases from 4° to 44°. Since we did not find this angle effect in Experiment 2 (angles ranging from 44° to 156°) or Experiment 3 (angles ranging from 16° to 178°), we assume that the SRM quickly becomes effective at small angles. Interestingly[51], reports that the minimum audible angle (MAA) is about 13.3° in crocodilians. This threshold suggests that the effect of angle on SRM may be dominant for small angles in the range 4°-15°, and becomes saturated for higher angular values.

Our results still support the hypothesis that SRM is a shared ability among vertebrates. Gray treefrogs showed SRM ranging from 3 to 12 dB[38,42]. Despite the enormous variability in SRM as a function of experimental context[22], birds develop high abilities to use spatial cues as a means to detect a target signal. For example, budgerigars *Melopsittacus undulatus* display a SRM of around 9 dB when required to detect pure tones against white noise in a Go/No-Go experimental setup[30], but achieve an impressive SRM of 20 to 30 dB in an identification task with biological signals[31]. In mammals, the SRM reaches 10 dB in ferrets[26], and 12 to 19 dB in pinnipeds[29]. In humans, the SRM has been estimated to be between 15 and 18 dB with "clicks" as target signals, the masker being broadband noise[24]. These high values could be explained by a greater ability to analyze auditory sound scenes, by different experimental conditions, or, in humans, simply by the fact that subjects are better able to understand the task required for the experiment. The amount of SRM measured in ethological studies is likely to be lower than in neurophysiological studies, due to perceptual and decision-

making effects. This makes it difficult to compare values found by an ethological approach such as the one employed here with values measured with a neurophysiological approach.

In conclusion, our several approaches–from field to laboratory experiments–demonstrate the use of SRM in crocodilians, and highlight that these amphibious animals can take advantage of the spatialization of sound sources in their natural environment to analyze sound scenes, and improve detection of signals containing relevant information. We argue that naturalistic approaches are absolutely necessary to fully understand and measure SRM abilities. For this and other biological processes, field experiments provide the ultimate proof of the relevance of a mechanism. The difficulty of conducting such experiments is offset by the naturalistic results they provide.

## Material and methods

### Experiment 1 (Field experiment): Spatial release from masking during mother-young communication in wild

*Field location and tested animals.* We conducted the field work at "Nhumirim ranch" (Embrapa Research Station, Mato Grosso do Sul, Brazil; 1859'16.1"S 5637'08.8"W), an area that covers 4310 ha with about 100 lakes[58,59]. We first surveyed the area for nests and Jacare caiman females in February–March 2019, and then conducted the playback experiments at the end of the hatching season (April 30th - May 11th 2019). We tested 16 adult females that had been previously identified as having built a nest and laid eggs. Most of the tested females were on separated lakes (10 of 16 individuals, Fig. 1a). When two females living in the same lake were tested successively (3 lakes × 2 individuals = 6 individuals) we always chose individuals separated by at least 100 meters, and carefully checked that the second female to be tested could not

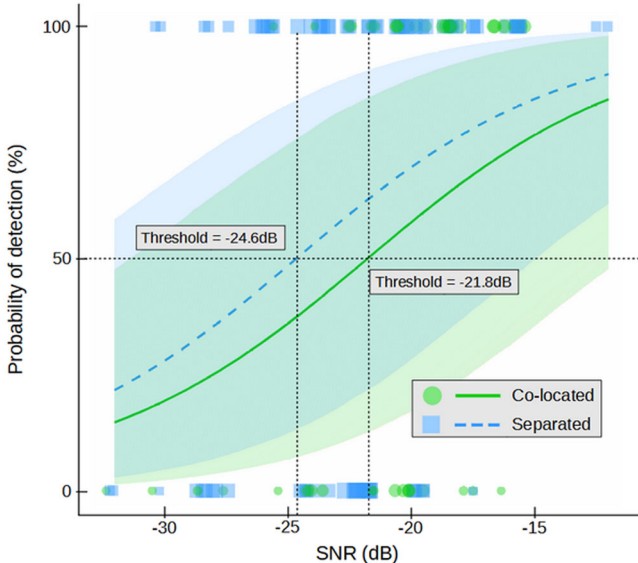

**Fig. 6 Effect of the Signal-to-Noise ratio (SNR) and of the loudspeakers' spacing ("co-located" versus "separated") on the behavioral reaction of juvenile Nile crocodiles to the playback of a synthetic buzz.** (Experiment 3, ENES Laboratory; the animals have been trained to move towards the target loudspeaker; curves = fitted probabilities of signal detection: mean of posterior distribution and 95% credible intervals; green dots represent individual trials in co-located condition, blue squares are individual trials in separated condition). The probability of target signal detection increases with SNR in both "co-located" and "separated" conditions. The tested crocodiles detect stimuli with lower SNR in the "separated" condition, supporting the hypothesis that they perform Spatial Release from Masking. The difference between both detection thresholds (amount of spatial release) is 2.8 dB.

have heard the sound stimuli broadcast to the first tested female. To avoid habituation, each female was involved in only one experimental session. All experiments were conducted during the day.

*Experimental signals.* We tested females with juvenile distress calls (Fig. 1c), which are well-known to elicit protective behavior from the mother[43]. The day before the first experimental session, we recorded distress calls from 3 Jacare juveniles approximately 3 weeks old. Calls were elicited by successively manipulating each individual. Handling time did not exceed 2-3 minutes and juveniles were immediately returned to their mother after being recorded. These individuals belonged to the same clutch, and their mother was not included in the females tested. Thus, the females tested were all tested with calls from juveniles that were not their own. Previous work has shown that female crocodilians respond indifferently to the calls of their young and the calls of unknown young[54,60].

During the playback experiments, we broadcast a "masking noise" and a "target signal". The masking noise was a white noise (2 hours duration, frequency range [20, 20000] Hz; 83 unweighted dB SPL measured at 1 m using a Sound level meter AMPROBE SM-10; slow time window equal to 1 s). It was broadcast in a loop for the duration of each experimental session. The target signals were designed as sequences of 10 successive distress calls (randomly selected from our bank of recorded calls). Each call was previously low-pass and high-pass filtered (cut-off frequencies: 20 Hz and 10 kHz respectively, 3rd order filters), and normalized in intensity by its RMS value (i.e. each call contained the same amount of energy). In each target signal, the duration of

silence between two calls varied randomly between 1.25 ± 0.25 s to reproduce a natural rhythm (total duration of the target signal = 17 s). We created 11 target signals, which differ from each other in their sound level. The intensity of the calls within each target signal was precisely adjusted to the intensity of the masking noise in the range [−20, 0] dB with a step size of 2 dB. The signal-to-noise ratios between the target signals and the masker were then computed directly from the intensities of the audio signals.

*Playback protocol.* Prior to an experiment, we placed three remote-controlled loudspeakers (FoxPro Fusion, rear loudspeaker, see Supplementary Fig. 6a, b for the technical specifications) just above the water surface, approximately 20 meters from the tested female (minimal distance = 12 m; maximal distance = 50 m; Fig. 1b). Two of the loudspeakers were placed side by side: one played the masking noise ("noise" loudspeaker), and the other was used to play back the target signal ("co-located" loudspeaker). The third loudspeaker ("separated" loudspeaker) was positioned to form an isosceles triangle with the noise loudspeaker and the initial position of the tested caiman female (Fig. 1b). This equidistance of the speakers from the female allowed us to consider the SNR value at the female's head position as equal to the SNR calculated at the speakers' emission. By estimating the distances between the speakers and the female, we calculated the separation angle θ as the angle formed by the female, the noise loudspeaker and the separated loudspeaker. Because the crocodiles were free to move, we could not ensure a constant angle between the female's head and the loudspeaker from trial to trial.

The target signals were alternately emitted from the co-located speaker and the separated speaker. At the beginning of the experiment, the female was at the same distance from the co-located loudspeaker and the separated loudspeaker (Fig. 1b). The masker was played continuously throughout the experimental session, starting with a quick fade-in until it raised to the intensity level of 83 unweighted dB SPL to avoid frightening the female with a sudden noise. We never noticed any change in the females' behavior during the 10 min after the masker appeared. Specifically, we did not notice any type of avoidance behavior of the loudspeaker emitting the masker.

Before playing back the first target signal we first observed the female's behavior for at least 5 min (Fig. 1d). If the female moved during this observation period, we waited another 5 min. If the female's distance from the co-located and separated loudspeakers was no longer equal, we then changed the position of the loudspeakers to recreate the isosceles triangle between the two loudspeakers (Fig. 1b), and we started another 5 min observation period before the experiment.

At the end of the observation period, we broadcast the first target signal from the co-located speaker at a low intensity level (SNR varying between −18 and −4 dB). The target signal was emitted 3 times, once per minute (Fig. 1d). However, the delay between these renditions was variable, depending on the female's behavior: if she moved or dived underwater, we waited for her to stop or to reappear at the surface before broadcasting the target signal again. After the third playback of the target signal, we waited at least 3 min, then repeated the same procedure this time from the separated loudspeaker (Fig. 1d). After a post-playback delay of at least 3 min, we would emit a new target signal increased by 2 dB, following the same procedure. An experimental session thus consisted of a repetition of this procedure, alternating the playback between the co-located speaker and the separate speaker, and increasing the signal-to-noise ratio by +2 dB in each cycle. The experimental session was stopped as soon as the female responded to the stimuli by orienting in the direction of the target speaker and/or approaching it. Specifically, we

stopped the playback when the female had changed her initial position by more than one body length. In summary, each female was tested with 1–9 pairs of target signals (each pair corresponding to a broadcast by the co-located speaker and a broadcast by the separated speaker).

*Analysis of behavioral reaction to playback.* We observed and filmed the behavior of the females throughout the experiments. Because the field experiments were conducted on wild animals with the ability to express their full range of behaviors, we assessed the response of the females by scoring their behavior as follows (motivation scale): Score 0: no behavioral response (no movement); Score 1: the female moved her head or body, but not in the direction of the target loudspeaker (misdirected response); Score 2: the female moved her head and looked towards the target loudspeaker without moving her body; Score 3: the female moved less than 1 body length towards the target loudspeaker; Score 4: the female moved more than 1 body length towards the target loudspeaker.

### Experiment 2 (experiment in zoo): Spatial release from masking during between-juveniles interactions

*Location and animals.* We performed these experiments in October 2019 at the "Crocoparc" zoo (Agadir, Morocco). We worked with naive juvenile Nile crocodiles (*Crocodylus niloticus*) hatched in captivity ($n = 8$ individuals; three months old; $36 \pm 2$ cm length). These animals were housed together in an exterior enclosure not visible by the public. They had never been included in any experiments before. Each crocodile subject was tested during only one experimental session.

*Experimental signals.* As in experiment 1, we broadcast masking noise and target signals. The masking noise (white noise, 2 h duration) was played continuously in a loop, starting before putting the crocodile in the pond and throughout each experimental session (frequency range [20, 20,000] Hz; 83 unweighted dB SPL at 1 meter with the same sound level meter and same settings as in experiment 1). As target signals, we used twelve different sequences of three identical Nile crocodile contact calls from our recording data bank (twelve unit calls from young Nile crocodile previously recorded in the Okavango Delta, Botswana, by T. Aubin and N. Mathevon; see spectrogram on Fig. 3b). Contact calls are known for maintaining cohesion among juveniles by soliciting their reunification[43]. Each call was previously band-passed filtered between 20 Hz and 10 kHz (filter order of 3), and its intensity was normalized by its RMS value. In each target signal, the duration of the silences between the calls was randomly set between 5 ± and 1.5 s (to match the natural rhythm), resulting in a total signal duration of 11 seconds. We adjusted the intensity level of the target signals (directly in the audio files, as in Experiment 1) to achieve an SNR in the range [−32, −16] dB with a 2 dB step.

*Playback protocol.* The experiments were performed outdoors at night in an artificial pond of approximately 40 m² (maximum dimensions: 6 × 7 meters; Fig. 3a). Four remotely controlled loudspeakers (FoxPro Fusion, rear loudspeaker, Supplementary Fig. 6a, b) were placed on the pond shore (Fig. 3a). As in experiment 1, two loudspeakers were placed side by side: one broadcasting the masking noise (noise loudspeaker) and the other emitting the target signal ("co-located" loudspeaker). The other two loudspeakers were placed at distance from the noise loudspeaker ("separated" loudspeakers; Fig. 3a). The location of the loudspeakers around the pond was changed between each tested subject to avoid positional bias and to cover a wide range of

angles between the target speaker, the noise speaker, and the crocodile's position. Given the size of the pond, the distance between the tested animal and the target loudspeaker was biologically relevant: in the wild, juveniles of the same groups are often one to a few meters apart. Prior to each trial, the tested juvenile was placed alone in the pond the afternoon before the playback of the target signals (at least 3 hours before dusk), allowing it to become accustomed to its new environment (Fig. 3c). The masking noise was broadcast continuously during this habituation period and throughout the experimental session. During the trials, the experimenters controlled the playback of the target signals while remaining 15m distant from the experimental pond, out of sight of the animal. The first target signal was broadcast by one of the randomly selected target loudspeakers (co-located or separated), at a random SNR value. If the crocodile had not moved 90 s after the end of the target signal played, the same signal was played again on the same loudspeaker (Fig. 3c). Then, we waited ten minutes after the last signal was played before playing another target signal (randomly chosen from the sound bank) from another target loudspeaker. On average, we performed 8.8 ± 1.4 trials per crocodile tested (Fig. 3c). Because the tested crocodile was free to move within the pond, its initial position varied between trials. Therefore, while the SNR value of the played back signal was chosen by the experimenters, the SNR actually perceived by the crocodile at the beginning of the playback depended on its position in the pond relative to the noise speaker and the target speakers. To measure the SNR perceived by the crocodile, we mapped the SNR variations at the pond surface by performing an acoustic propagation experiment. For this propagation experiment, we played back distress calls and noise, and measured their intensity at different points in the pond. This allowed us to model an acoustic map of the pond representing the intensity variations as a function of the position of the crocodile in the pond (Fig. 7). The SNR corresponding to the position of each crocodile tested was then calculated from the intensity levels of the target signal ($L_T$) and masker ($L_M$) and the position of the animal. For each experiment, the initial perceived SNR ($SNR_p$) was defined as follows:

$$SNR_p(dB) = L_T - L_M. \tag{1}$$

For each trial, we also measured the separation angle $\theta$ formed by the crocodile, the noise loudspeaker and the target loudspeaker. This angle was constrained both by the experimental configuration (Fig. 3a) and by the initial position of the tested juvenile, and varied between 44 and 156°. As in Experiment 1, we were unable to ensure a constant angle between the crocodile's head and the masker and/or the target from trial to trial.

*Analysis of behavioral reaction to playback.* We observed and filmed the behavior of the juveniles during all the trials (infrared cameras ABUS TVCC34010). The videos were analyzed using Kinovea software (www.kinovea.org). In order to accurately measure the positions in the field and the distances traveled by the crocodiles, we took care to correct the distortion of the camera lens and the geometric perspective error. We extracted the position coordinates of the loudspeakers and the crocodile (the point between the eyes) at the beginning of each playback. Based on these coordinates, we calculated the separation angle $\theta$ between the crocodile, the noise loudspeaker, and the target loudspeaker.

To assess the juvenile's response to the target signal, we used a binary scale (detection scale), giving a score of 1 if the juvenile showed significant orientation or movement toward the target loudspeaker and 0 if it still had not responded at the end of the

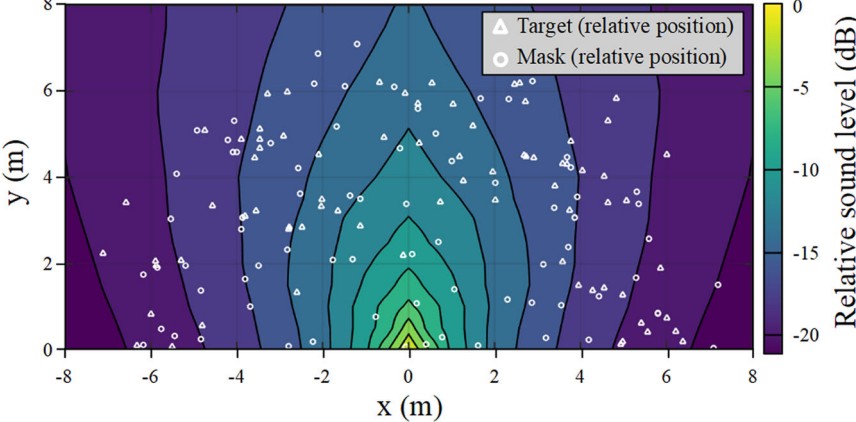

**Fig. 7 Acoustic propagation of a juvenile call and white noise on the pond.** The position of the sound source is normalized at (0, 0). The white triangles and circles represent respectively the positions of the crocodile relatively to the target (triangles) and noise (circles) loudspeakers at the beginning of each experimental trial. The sound intensity level is coded by the color scale.

playback of the target signal. We also measured the reaction time (in seconds) between the first observed behavioral response and the stimulus onset.

### Experiment 3 (Go/No-Go experiment in the laboratory): spatial release from masking with trained animals

*Location and animals.* We conducted these experiments between March and June 2019 at the ENES laboratory. We worked with two Nile crocodiles born in captivity at the zoo "La Ferme aux Crocodiles" (Pierrelatte, France). These animals were three years old (biometric data are available in the Supplementary Table 14) and they were housed at the ENES animal facilities. They had been previously included in an experiment on sound localization[51], involving a conditioning procedure using harmonic complex tones (buzz). In this experiment, each subject was tested once a week for 14 weeks.

*Experimental signals.* As in Experiments 1 and 2, we broadcast a masking noise and target signals. The masking noise (white noise, 2 hours duration) was played continuously in a loop before the tested subject was placed in the experimental room and throughout each trial (frequency range [20, 20000] Hz; 60 dB SPL at 50 cm). The target signals were sequences of three different synthetic buzzes (harmonic complex tones; fundamental frequency $f_0 = 208$, 220 and 233 Hz; duration = 500 ms each; signals synthesized with Python 3.7, SciPy; Fig. 5b). Each target signal was designed as a repetition of three identical buzzes, separated by an interval of $2 s \pm 500$ ms (total duration of each target signal = 9 s). The intensity level of the target audio signals was adjusted to achieve a signal-to-noise ratio in the range [−32, −16] dB with a 2 dB step, as in experiments 1 and 2.

*Behavioral conditioning.* Prior to the experiment, the two Nile Crocodiles were trained twice a week to come towards a sound source. The training followed a classical Go/No-Go procedure[51]: two speakers were placed at the edge of the pool, with only one emitting target signals. As soon as the crocodile touched the target speaker with its snout, it was rewarded with a piece of meat. Before and after the test period, both individuals achieved 100% success in the conditioning sessions.

*Playback protocol.* The experiments were conducted in the dark in a dedicated sound attenuation chamber (TipTopWood©, dimensions = 1.8 × 2.3 × 2.2 meters, background noise <40 dB SPL, reverberation time = 0.44 s; Fig. 5a), where a squared pool

(1.75 m wide) had been set up for the purpose of the experiment. The pool was filled with water to a level that allowed the crocodile to swim (water depth = 10 cm[50]). Four loudspeakers (AudioPro, Bravo Allroom Sat, Supplementary Fig. 6c, d) were installed just beyond the water surface at the edge of the pool (Fig. 5a). During each trial, a loudspeaker continuously broadcast the masking noise ("noise" loudspeaker). In the co-located condition, the same speaker also played the target signal mixed with the noise. For the separate condition, two speakers placed at different locations could play the target signal. To maintain the motivation of the crocodiles to respond to the target signals, we chose to reward them each time they came to the target speaker during the experimental trials. It was indeed not possible to reinforce the behavioral response of these animals outside of the experiments if we wanted the animals to maintain their motivation to respond to the signals. In front of each speaker was a system that hid food (a small piece of meat) to reward the animal if it approached the target speaker in response to the stimulus. To control for the possible effect of the smell of the food, we placed a fourth speaker, always silent, accompanied like the other three by the system hiding the food (but which was never delivered to the animal). This loudspeaker was never approached in response to a sound. With the exception of the co-located / noise loudspeaker, the spatial locations of the loudspeakers were changed between each experimental session. The sound emission chain consisted in two computers and two power amplifiers (Yamaha AX-397) connected to the loudspeakers and placed outside the experimental chamber. We recorded the behavior of the tested subject with an infrared camera (ABUS TVCC34010) connected to a computer. The tested crocodile was released into the pool at least 20 min before the start of an experimental session (Fig. 5c). The noise loudspeaker was already on and was not turned off until the end of the experimental session (Fig. 5c). The first target signal was broadcast either from a separated loudspeaker or from the co-located loudspeaker at a specific intensity level (both parameters were randomly picked). If the crocodile had still not moved 45 s after the third buzz of the target signal ended, we repeated the same target signal once (Fig. 5c). The crocodile was rewarded if it approached the target loudspeaker within 5 min of the last buzz. If the crocodile responded correctly (movement toward the target loudspeaker), we waited 5 min before starting another trial. On average, we performed 9 ± 2 trials during an experimental session, covering a wide range of SNRs. Each session always included a few trials at high SNR to check the crocodile's motivation to respond. The crocodile was then left 20 min in the pool before

being recaptured (Fig. 5c), to limit an association between the final target signal and a stress-inducing event. We measured the separation angle $\theta$ (the angle formed by the crocodile, the noise loudspeaker and the target loudspeaker) at the onset of the playback. This angle was constrained both by the configuration of the experimental set-up (Fig. 5a), and the initial positions of the tested subject, and varied between 16 and 178°. As for the two other experiments, we could not ensure a constant angle between the crocodile's head and the masker and/or target loudspeaker from trial to trial.

*Analysis of behavioral reaction to playback.* As in Experiments 1 and 2, we observed and filmed the behavior of the tested subjects throughout the experiments. Video analyses were performed using Kinovea software. Before video analysis, we corrected for camera lens distortion and geometric perspective error. We measured the position coordinates of the loudspeakers and the initial positions of the crocodile (using the point between the eyes) before the start of the playback. Based on these coordinates, we calculated the separation angle $\theta$ between the noise loudspeaker, the crocodile and the target loudspeaker.

To assess the tested subject's ability to detect the target sound against the background noise, we used the same binary scale as in the second experiment (detection score), giving a score of 1 if the juvenile showed orientation or movement toward the target loudspeaker, and 0 if it still did not respond within the 5-min observation period following the last buzz of the target signal. If the crocodile did not respond more than twice to one of the higher SNRs (−16 dB or −18 dB) in the same session, the entire session was excluded from the final data set, considering that the motivation to respond to the target signal was not sufficient (only one session had to be excluded). We also measured the latency to respond, i.e. the time between the animal's first response and the preceding target signal.

**Statistics and reproducibility**. All statistical analyses were performed in R (v.3.6.2) from a Bayesian perspective, which provides more flexible and considerably richer investigations than the frequentist approach. Bayesian algorithms also have the advantage to be robust for any sample size. Each of the behavioral responses was investigated using this approach: the behavioral response ([0–4]; Experiment 1) score or the signal detection (0 or 1; Experiments 2 and 3), and the latency time to react. These variables were modeled using Bayesian mixed models with random intercepts per tested crocodiles, fit in Stan computational framework accessed with brms package[61]. Behavioral scores were modeled with a cumulative link function, a powerful model that is too often underestimated and left out for ratings data (ordinal regression)[62]. The most appropriate link function was chosen by selecting the most predictive models. Two independent cumulative models were constructed as follows: a first one with the SNR and the position of target source (i.e. either co-located or separated) as fixed factors, and a second one by focusing only on the data obtained in the separated condition, with the SNR and the initial angle of separation between the target and the masker as fixed factors. Detection scores were modeled using the Bernoulli distribution (logistic regression), with SNR and position of the target source (co-located or separated) as fixed factors. This first model allowed us to approximate the signal detection threshold T50 (corresponding to a 50% probability of target detection[63] in both the co-located and the separated condition. As in the first experiment, a second model was based only on the data obtained in the separated condition and included the SNR and the initial angle of separation as fixed factors. Finally, when signals were detected, the effect of SNR and the position of the target source

on the animals' reaction time was investigated using a skewed normal distribution to consider its asymmetry. All models were based on four chains of 10000 iterations with 2000 warmup samples. Model convergence was checked with traceplots and the Gelman-Rubin's potential scale reduction factor (R̂ equal to 1.00,[64]) on split chains. The interaction between fixed factors was tested and removed from each model, as it reduced the fit of the models (WAIC calculated based on the posterior likelihood,[65]). The regression coefficients for each model were summarized using the mean of their posterior distribution and the 95% credible interval, reported in the text as 95% CI. Contrasts were reported using the median of the posterior distributions and the 95% credible intervals.

**Reporting summary**. Further information on research design is available in the Nature Research Reporting Summary linked to this article.

## Data availability
Dataset, codes, videos, audio signals and supplementary information supporting the present results can be found in the Zenodo repository[66] https://doi.org/10.5281/zenodo.5971364.

## Code availability
R codes used to generate all the results in this paper can be found at[66] https://doi.org/10.5281/zenodo.5971364.

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

## Acknowledgements

The authors are grateful to "Crocoparc" zoo in Agadir, Morocco (Luc Fougeirol, Patrice Brunet, Ariane Marinetti, Leila Sdigui, Philippe and Christine Alleon and the staff), "La ferme aux crocodiles" zoo, Colleen Reichmuth and Caroline Casey, and the EMBRAPA technical team (Luis Alberto Rondon and Denis Celin Tilcara). This research has been funded by the National Geographic Society (field experiments), the Institut universitaire de France (NM), the Labex CeLyA (PhD fundings to Leo Papet and Julie Thévenet, Lyon Center of Acoustics ANR-10-LABX-60), the CNRS and the University of Saint-Etienne.

## Author contributions

J.T., L.P., N.B., N.G., and N.M. conceived the experiments, J.T., L.P., N.B., Z.C., N.G., and N.M. conducted the experiments, J.T., L.P., N.G., and N.M. analyzed the results, J.T., and L.P. wrote the original draft, J.T., L.P., N.G., N.M., and M.G. reviewed and edited the manuscript.

## Competing interests

The authors declare no competing interests.

## Ethical approval

All experiments were performed in accordance with relevant guidelines and regulations including French and Brazilian national guidelines, permits and regulations regarding animal care and experimental use (approval no. D42-218-0901, ENES lab agreement, Direction Départementale de la Protection des Populations, Préfecture du Rhône).
