## [Peer Review File · Communications Biology]

Reviewers' comments:

Reviewer #1 (Remarks to the Author):

The study on crocodylian presents data on the improvement of signal detection by spatially separating sound sources producing target signals of interest and acoustic maskers, i.e., investigates spatial release from masking. It combines data from three different approaches in two different species: (1) a field study on caimans in the Pantanal exploiting a natural response of females to distress calls of young, (2) a study on young Nile crocodiles in a zoo exploiting their response to conspecific contact calls, and (3) a laboratory study on juvenile Nile crocodiles that are trained to report signal detection in a Go/NoGo operant procedure. All three studies demonstrate that crocodylians experience spatial release from masking that has so far been demonstrated in a number of species.

However, the study has also some weaknesses limiting the conclusions that must be noted. First, since habituation is a problem for the study parts (1) and (2), data could only be obtained from a limited number of trials. Second, the head position of the animal relative to the sound source positions and their angular separation as seen from the position of the listening crocodile varied in all three experiments. Thus, not many data points could be obtained for a specific angular separation of the sound sources. Finally, in all three experiments, there was a considerable variation in the signal-to-noise ratio that enabled a response (e.g., see supplementary figure 1). Taken together, these factors cause the estimate of the spatial release from masking to become quite inaccurate. Thus, it is not clear if the numerical values given for the three different experiments are representative. The three values for the amount of spatial release from masking all suggest that there is such a perceptual effect. However, its size cannot be accurately determined making it difficult to compare the present result with those from previous studies in other species. In addition, the Bayesian statistical approach make it difficult comparing the variability if the present results with those in previous studies. In summary, I do not want to say that the authors did not try their best, but the limitations of the experiments did not allow more accurate results.

Furthermore, it would have been nice linking the results of the present study with results from more studies investigating the binaural processing in crocodylians. For example, the studies from the group of Catherine Carr at the university of Maryland on alligators should be mentioned (e.g., Carr, C. E., & Christensen-Dalsgaard, J. (2015). Sound localization strategies in three predators. *Brain, behavior and evolution*, 86(1), 17-27 and other more recent studies from the Carr lab).

For the calculation of the spatial release from masking it is not necessary that the signal-to-noise ratio is adequately calculated, it only must be calculated similarly for the collocated and separated loudspeaker positions. However, it would have been nice to find better estimates for the signal-to-noise ratio at threshold in the present study. The A-weighting function used in the present study to calculate the signal-to-noise ratios is more or less adequate for human perception, but unlikely adequate for signal detection in noise by crocodylians. Crocodylians have much narrower frequency range of hearing than humans. If this would be taken into account, better estimates of the signal-to-noise ratio at the detection threshold would have been obtained.

Reviewer #2 (Remarks to the Author):

The experiments conducted here were extremely well thought out and conducted, and represent a best case scenario of combining an effect found in the laboratory and measured in controlled captive conditions with demonstrating the phenomenon in captive conditions as well as wild conditions. The authors took care to calibrate their sound fields as best as they could, while still allowing the animals to roam freely during the trials. Thus, their measurements were not exactly controlled with equal trials across condition types, but represent natural behaviors of these animals. The results appear sound

across experiments, and my main complaints have to do with a lack of description about trials and some wording issues.

I found myself wondering about various factors in this experiment such as the problem of the 'null-angled' condition occurring at different locations around the crocodiles across trials, the differences across stimulus types in the three experiments, the unnatural reverberation in the third experiment, and the differences in MAA and SRM at different points around the crocodiles' heads. I had to delete my questions one by one as I read through the discussion, as all were addressed in a thoughtful manner. The authors acknowledge that there are methodological issues that they could not control. I agree with these issues, but also agree that the foundations of the SRM effect are shown here, for the first time, despite those issues. I think that had you been able to always present the masker at 0 degree azimuth, your effect would have been much larger across experiments.

Minor points:

Could you please describe the evolutionary relationship of Jacare caimans and Nile crocodiles in the intro?

Line 25 – not sure what 'null-angled' means – perhaps define as 'spatially coincident'. To me, null implies directly in front of the receiver, always at a 0 degree angle. This is not how it is used in this paper.

Line 28 – missing comma after 27

Second paragraph of introduction – Feng and Ratnam also investigated aspects of frog hearing in the wild in the presence of noise, as well as neural bases of SRM.

Line 41 – change 'and have a limited' to 'with a limited'

Line 46 – change 'combined to' to 'combined with'

Line 66 – add an 's' to the end of 'remove'

Fig 1 caption – add an 's' to the end of loudspeaker in the last line. Also please specify the solid areas in the green and blue blocks are stimulus playback and the dashed lighter areas immediately following the playback are the recording blocks.

Did you always test coincident first and then separated second? Or was it random across subjects? If the former, do you think there are any implications? E.g., what if animals 'thought' they detected something but it was hard to hear because coincident, then 'definitely' detected something when separated because of SRM – so their responses were always more robust for the segregated speaker trials? If randomized, please say so.

Fig 3 caption – put an apostrophe at the end of loudspeakers in the title (or delete the s). Also, this is the third location (so far) where you've described what the 0-4 scale represents. You can delete here at least. You do the same in experiment 2 (delete in fig 6 caption).

In fig 3B – I'm really surprised the errors aren't larger for the separated condition considering they cover a very large 40 degrees of separation conditions. In SRM, there is usually a minimum separation that works to improve thresholds – but then increasing separations increase the masking release (as you show in 3c). If this was the case, your variability should be larger here, no?

Fig 3C – I'd delete the white box in the habituation section of the pink bar. It suggests you turn the noise off. You could instead put slashes representing an extended habituation time.

Last line of Fig 3 caption – 'behavioral reaction increases when separation angle increases'. This is only true for the 4th response type. Reactions actually decrease after a certain angle in 2 and 1, and they hold flat for 3. Why might this be happening?

Figure 4 – I'm confused about the 1&2 on the timeline. Do you alternate between the two loudspeakers? If so, maybe put '1' and '2' next to the two blue speakers for clarity – and also mention them in the description of 4C. Alternately, if you are mentioning two playbacks from the same loudspeaker, please state that. Either way, did you always do left then right (or vice versa) or did the conditions differ across animals? Did you always do separated first, then coincident? Or did you randomize across subjects? Please state in methods.

Fig 6 caption, line 4 – I'd delete 'higher' behavioral response here. It isn't like experiment 1 where you rank responses. Here they just elicited any response. You can say probability of detection increases with SNR, which is accurate. Also in this figure – do the dots and squares along the top and bottom represent individual trials? Subjects?

Line 179, change 'for' to 'in'

Figure 9 caption – again put an apostrophe at the end of loudspeakers or delete the `s`. Again, please describe dots and squares along the top and bottom of the figure.

Line 287 – was the white noise a looping frozen sample? A non-repeating stream? Please specify.

Line 298 – what were the frequency characteristics of the speakers?

Line 310 – change `mask` to `masker` (and in other instances where you wrote mask), add a `to` in after `raised`

Line 430 – how many sessions did you have to exclude for this reason?

Response to Referees

Reviewer #1:

The study on crocodylian presents data on the improvement of signal detection by spatially separating sound sources producing target signals of interest and acoustic maskers, i.e., investigates spatial release from masking. It combines data from three different approaches in two different species: (1) a field study on caimans in the Pantanal exploiting a natural response of females to distress calls of young, (2) a study on young Nile crocodiles in a zoo exploiting their response to conspecific contact calls, and (3) a laboratory study on juvenile Nile crocodiles that are trained to report signal detection in a Go/NoGo operant procedure. All three studies demonstrate that crocodylians experience spatial release from masking that has so far been demonstrated in a number of species.

We are grateful to the referee for their accurate analysis of our work and their comments. We did our best to take into consideration all of their questions and requests.

However, the study has also some weaknesses limiting the conclusions that must be noted. First, since habituation is a problem for the study parts (1) and (2), data could only be obtained from a limited number of trials. Second, the head position of the animal relative to the sound source positions and their angular separation as seen from the position of the listening crocodile varied in all three experiments. Thus, not many data points could be obtained for a specific angular separation of the sound sources. Finally, in all three experiments, there was a considerable variation in the signal-to-noise ratio that enabled a response (e.g., see supplementary figure 1). Taken together, these factors cause the estimate of the spatial release from masking to become quite inaccurate. Thus, it is not clear if the numerical values given for the three different experiments are representative. The three values for the amount of spatial release from masking all suggest that there is such a perceptual effect. However, its size cannot be accurately determined making it difficult to compare the present result with those from previous studies in other species. In addition, the Bayesian statistical approach make it difficult comparing the variability if the present results with those in previous studies. In summary, I do not want to say that the authors did not try their best, but the limitations of the experiments did not allow more accurate results.

Habituation is indeed a major problem when testing crocodiles. These animals are particularly smart (they are top predators) and learn very quickly that the sound stimuli sent are not associated with the actual presence of an individual. This rapid habituation forced us to limit considerably the number of trials and replicates during the experiments conducted in the field in Brazil and in semi-captivity (zoo). For the Go/No-Go experiments in the laboratory, we had to face two other constraints: 1) the limited number of individuals at our disposal (we had only 2 crocodiles at our disposal – discussed lines 202-204), 2) the limited motivation of the animals

to respond to the stimuli: crocodiles are ectotherms and can go without food for days or even weeks. This limits the number of successive trials that can be performed and imposes a long delay between two experiments for the animal to become hungry again (now discussed lines 204-206). In the field, the solution we chose was to present successively the signals starting from the lowest SNR until the one inducing a response from the animal. We are aware that this way of doing things can create a cumulative effect, but we were very careful to leave a significant temporal delay between the stimuli to limit it. At the zoo and in the lab, we optimized the number of stimuli that could be tested in a session without increasing habituation too quickly by placing several speakers around the ponds in order to change the origin of the sound (now discussed lines 201-202). This device allowed us to play a larger number of stimuli. Still, being aware of the limited number of trials per experiments, Bayesian models have this advantage to be robust and appropriate even for small sample sizes, which is not the case of classical frequentist analysis. The justification for the choice of the statistical method was added in methods, line 484: “Bayesian algorithms also have the advantage to be robust for any sample size.”

We chose to investigate SRM with freely-moving animals, which meant that the position of the head when listening to the stimuli varied from one experimental trial to another (discussed lines 255-259). We also had constraints on the positioning of the speakers. For example, in the field it was not possible to position the speaker far from the shore (for safety reasons), which limited the amplitude of the possible spacing between the speakers. However, considering all the experiments we conducted, we were still able to cover a wide range of angles.

As the referee points out, we observed considerable variation in SNR inducing crocodile response. This was expected in field or semi-captive experiments, as individual variations in animal motivation and personality easily explain different response thresholds. For example, the females tested in the field in the Pantanal were with their young: it is possible that a female is reluctant to abandon them to respond to the call of a lone young (which we mimicked with our playback cues), and that this reluctance varies from one female to another (discussed lines 221-226). In the zoo, being alone in the basin could be stressful for the young tested, with possibly varying consequences depending on the personality of the individuals (discussed lines 227-229). Certainly, we expected to see less inter-individual variability in the Go/No-Go experiments in the laboratory since animals were trained to respond to stimuli. However, here too, we observed variations, potentially related to personality differences (one animal was clearly bolder than the other, and responded more readily to stimuli) (now discussed lines 231-233). As the referee points out, these variations make it difficult to measure absolute and exact SRM values, as indicated lines 212-216. However, they reflect the reality of the animals' behavior and, despite them, we were able to establish that crocodiles use SRM and give an order of magnitude of its value. Finally, we agree with the reviewer that its effect size cannot be accurately determined. This is the reason why we only claim in the conclusions (and in the abstract) that the effect of SRM is maximum (saturated) over 15°. We also agree that this makes difficult to compare this result with results from others studies with other species, as now discussed lines 278-280.

Furthermore, it would have been nice linking the results of the present study with results from more studies investigating the binaural processing in crocodylians. For example, the studies from the group of Catherine Carr at the university of Maryland on alligators should be mentioned (e.g., Carr, C. E., & Christensen-Dalsgaard, J. (2015). Sound localization strategies in three predators. *Brain, behavior and evolution*, 86(1), 17-27 and other more recent studies from the Carr lab).

Thanks for the referee for pointing this out. We have now added and discussed these studies lines 251-255:

“In our study, a reliable SRM effect was reported in all experiments, and all three mechanisms described in human could also have contributed to the observed SRM effect. In particular, it is now well known that crocodile ears are acoustically coupled by air-filled cranial sinuses^{21,49}, which greatly increases directional cues such as ITDs⁵⁷. At the encoding level, alligators have been shown to form ITD maps in the brainstem nucleus laminaris similarly to birds, again suggesting a convergence among modern archosaurs⁵³. The crocodylian binaural system may therefore be as well-developed as that of birds and thus could be effective in detecting spatially separate signals⁵².

21. Carr, C. E., Christensen-Dalsgaard, J. & Bierman, H. *Coupled ears in lizards and crocodylians. Biol. Cybern.* 110, 291–302, [10.1007/s00422-016-0698-2](https://doi.org/10.1007/s00422-016-0698-2) (2016).

52. Bierman, H. S. & Carr, C. E. *Sound localization in the alligator. Hear. Res.* 329, 11–20, [10.1016/j.heares.2015.05.009](https://doi.org/10.1016/j.heares.2015.05.009) (2015).

53. Kettler, L. & Carr, C. E. *Neural maps of interaural time difference in the american alligator: a stable feature in modern archosaurs. J. Neurosci.* 39, 3882–3896 (2019).

57. Carr, C. E. & Christensen-Dalsgaard, J. *Sound localization strategies in three predators. Brain, Behav. Evol.* 86, 17–27, [10.1159/000435946](https://doi.org/10.1159/000435946) (2015).

For the calculation of the spatial release from masking it is not necessary that the signal-to-noise ratio is adequately calculated, it only must be calculated similarly for the collocated and separated loudspeaker positions. However, it would have been nice to find better estimates for the signal-to-noise ratio at threshold in the present study. The A-weighting function used in the present study to calculate the signal-to-noise ratios is more or less adequate for human perception, but unlikely adequate for signal detection in noise by crocodylians. Crocodylians have much narrower frequency range of hearing than humans. If this would be taken into account, better estimates of the signal-to-noise ratio at the detection threshold would have been obtained.

Regarding the sound levels in A-weighted decibels, we agree with the referee. We now give the sound level values in unweighted dB SPL which are more appropriate:

- Line 82: “For each female (N = 16), we played back a broadband noise ("noise source", emitted at 83 dB SPL, unweighted) simultaneously with a series of distress calls recorded from young juveniles”

- Line 308: “The masking noise was a white noise (2 hours duration, frequency range [20, 20000] Hz; 83 unweighted dB SPL measured at 1 m using a Sound level meter AMPROBE SM-10; slow time window equal to 1 second).”

- Line 332: “The masker was played continuously throughout the experimental session, starting with a quick fade-in until it raised to the intensity level of 83 unweighted dB SPL to avoid frightening the female with a sudden noise.”

- Line 367: “(frequency range [20, 20000] Hz; 83 unweighted dB SPL at 1 meter with the same sound level meter and same settings as in experiment 1).”

In our study, the calculation of the signal-to-noise ratios is independent of any weighting function. The signal-to-noise ratio corresponds to the energy difference between the target signal and the masker (in the field: both loudspeakers are at the same distance from the tested female; in the laboratory: the sound level was constant over the entire surface of the pool). This difference was calculated directly from the broadband audio signals (without weighting). Thus, the signal-to-noise ratio values presented in the manuscript were not affected by any weighting function and were calculated similarly regardless of the target configuration (co-located or separated). We now clarify this point in the manuscript:

- Lines 316-317: “The signal-to-noise ratios between the target signals and the masker were then computed directly from the intensities of the audio signals.”

- Line 374: “We adjusted the intensity level of the target signals (directly in the audio files, as in Experiment 1) to achieve an SNR in the range [-32, -16] dB with a 2 dB step.”

- Lines 431-432: “The intensity level of the target audio signals was adjusted to achieve a signal-to-noise ratio in the range [-32, -16] dB with a 2 dB step, as in experiments 1 and 2.”

Reviewer #2:

The experiments conducted here were extremely well thought out and conducted, and represent a best case scenario of combining an effect found in the laboratory and measured in controlled captive conditions with demonstrating the phenomenon in captive conditions as well as wild conditions. The authors took care to calibrate their sound fields as best as they could, while still allowing the animals to roam freely during the trials. Thus, their measurements were not exactly controlled with equal trials across condition types, but represent natural behaviors of these animals. The results appear sound across experiments, and my main complaints have to do with a lack of description about trials and some wording issues.

I found myself wondering about various factors in this experiment such as the problem of the ‘null-angled’ condition occurring at different locations around the crocodiles across trials, the differences across stimulus types in the three experiments, the unnatural reverberation in the third experiment, and the differences in MAA and SRM at different points around the crocodiles’ heads. I had to delete my questions one by one as I read through the discussion, as all were addressed in a thoughtful manner. The authors acknowledge that there are methodological issues that they could not control. I agree with these issues, but also agree that the foundations of the SRM effect are shown here, for the first time, despite those issues. I think that had you been able to always present the masker at 0 degree azimuth, your effect would have been much larger across experiments.

We are very grateful to the reviewer for their kind and helpful comments. We greatly appreciate their understanding of the difficulty of experimenting with wild animals in the field. We have taken all of their comments into account.

Minor points:

Could you please describe the evolutionary relationship of Jacare caimans and Nile crocodiles in the intro?

Added lines 73-74:

“Jacare caimans and Nile crocodiles are two representatives of two of the three extant groups of crocodylians that differentiated during the Cretaceous: the Alligatoroidea and Crocodyloidea respectively (the third group being the Gavialoidea).”

Line 25 – not sure what ‘null-angled’ means – perhaps define as ‘spatially coincident’. To me, null implies directly in front of the receiver, always at a 0 degree angle. This is not how it is used in this paper.

We agree with the referee that the term “null-angled” can be confusing. We follow their suggestion and have replaced this term by “co-located” throughout the manuscript and figures.

Line 28 – missing comma after 27

Done.

Second paragraph of introduction – Feng and Ratnam also investigated aspects of frog hearing in the wild in the presence of noise, as well as neural bases of SRM.

We thank the reviewer for pointing out this work to our attention. We now report this study in the introduction:

Lines 31-33: “SRM has been investigated in amphibians (northern leopard frogs *Rana pipiens pipiens*³², Cope's gray treefrog *Hyla chrysoscelis*³³⁻³⁷), showing improved detection and discrimination of conspecific calls masked by noise when the two sources are spatially separated^{22,38}.”

32. Ratnam, R. & Feng, A. *Detection of auditory signals by frog inferior collicular neurons in the presence of spatially separated noise. J. Neurophysiol. 80, 2848–2859 (1998).*

Line 41 – change ‘and have a limited’ to ‘with a limited’

Done.

Line 46 – change ‘combined to’ to ‘combined with’

Done.

Line 66 – add an ‘s’ to the end of ‘remove’

Done.

Fig 1 caption – add an ‘s’ to the end of loudspeaker in the last line. Also please specify the solid areas in the green and blue blocks are stimulus playback and the dashed lighter areas immediately following the playback are the recording blocks.

We have clarified the Figure legend as follows (also for Figure 4 and Figure 7):

“The masking noise is emitted continuously (red solid line). The target signals (with different SNR) are emitted either by the co-located loudspeaker (green solid line) or by one of the separated loudspeakers (blue solid line). (...) The light dashed lines represent the behaviour recording following the target emission.”

Did you always test coincident first and then separated second? Or was it random across subjects? If the former, do you think there are any implications? E.g., what if animals ‘thought’ they detected something but it was hard to hear because coincident, then ‘definitely’ detected something when separated because of SRM – so their responses were always more robust for the segregated speaker trials? If randomized, please say so.

As explained in the manuscript, we indeed always tested the co-located condition first and then the separated condition for the same SNR. The referee points out the possibility of a kind of "cumulative" effect where the female could have (mis)heard the signal in the localized condition and responded only in the separate condition. Despite this risk, we decided to follow this protocol to ensure as much as possible that each female could be tested in both conditions for each SNR level. If both conditions had been presented randomly, we would have had females approaching the speaker emitting the delocalized signal before the localized signal could be played, and it would then have been impossible to test them with that signal (the position of the female near the delocalized speaker would have prohibited any further playback). However, we took a number of precautions. First, we took into account the smallest observable behavioral response suggesting a possible detection of the signal (head orientation). In addition, we repeated the same reading 3 times, with fairly long intervals of 1 minute. If a female detected stimulus but hesitated to respond at the end of the first broadcast, she had two more opportunities to do so. Finally, we left three-minute intervals between the co-located and separate playback. This protocol led to rather clear responses from the females: when she responded to either cue with an orientation, it was always in the direction from which the cue came.

We now report this point in the discussion Lines 195-200: “In the field, to ensure as much as possible that each female could be tested in both co-located and separated conditions for several SNR, we choose to present successively the signals starting from the lowest SNR until it elicits a response from the animal. To avoid a potential cumulative effect due to this protocol, we took several precautions: (1) we were very careful to note the smallest observable behavioural response suggesting a possible detection of the signal, (2) we repeated successively the same signal 3 times to allow the female the opportunity to respond when they hesitated, and (3) we leave a significant temporal delay between the stimuli.”

Fig 3 caption – put an apostrophe at the end of loudspeakers in the title (or delete the s).

Done.

Also, this is the third location (so far) where you’ve described what the 0-4 scale represents. You can delete here at least. You do the same in experiment 2 (delete in fig 6 caption).

As suggested by the referee, we have removed redundant information in the Figures caption.

In fig 3B – I’m really surprised the errors aren’t larger for the separated condition considering they cover a very large 40 degrees of separation conditions. In SRM, there is usually a minimum separation that works to improve thresholds – but then increasing separations increase the masking release (as you show in 3c). If this was the case, your variability should be larger here, no?

As assessed by the Bayesian model, the figure here indicates that the strength of the response to the target signal increases with SNR, independently of the degree of separation between the target and the masker. Considering the data and the model, we are strongly confident that the SNR has an effect on signal detection regardless of the angle considered. The figure simply illustrates this: rather than showing the dispersion of the raw data, the credible intervals displayed in the figure show the dispersion of the posterior probabilities. This explicitly indicates the uncertainty of the SNR effect. Specifically, Bayesian algorithms allows us to know the uncertainty in the parameter estimate, and we then use all credible values of that estimate to derive the fitted probabilities.

To clarify the difference between the raw data distribution and the posterior distribution derived from the Bayesian model, we have now added that the raw data can be seen in Supplementary Figure 1 when quoted, and also for all experiments:

Figure 3: “The probability of eliciting a higher behavioral response increases with SNR (fitted probabilities of behavioral scores: mean of posterior distribution and 95% credible intervals).”

Figure 6: “(Experiment 2, Crocoparc Zoo, freely moving animals in a large basin; curves = fitted probabilities of behavioral scores: mean of posterior distribution and 95% credible intervals; green dots represent individual trials in co-located condition, blue squares are the individual trials in separated condition).”

Figure 9: “(Experiment 3, ENES Laboratory; the animals have been trained to move towards the target loudspeaker; curves = fitted probabilities of signal detection: mean of posterior distribution and 95% credible intervals; green dots represent individual trials in co-located condition, blue squares are the individual trials in separated condition).”

Fig 3C – I’d delete the white box in the habituation section of the pink bar. It suggests you turn the noise off. You could instead put slashes representing an extended habituation time.

Done.

Last line of Fig 3 caption – ‘behavioral reaction increases when separation angle increases’. This is only true for the 4th response type. Reactions actually decrease after a certain angle in 2 and 1, and they hold flat for 3. Why might this be happening?

Females showed rapid interest in the signals as the angle of separation increased, as evidenced by the decrease in scores of 0 and 1. Their first notable response was a head or body orientation,

hence the increase in score of 2 concomitant with the decrease in scores of 0 and 1. Then, for even larger angles, females showed stronger behaviour by approaching the loudspeaker (score of 4), rather than simply displaying a head or body orientation. Hence the decrease in the probability of getting a score of 2 after a certain angle. The probability of getting a score of 3 remained stable and low, indicating that females rarely move a little closer to the loudspeaker: once they have decided to respond to the signal, they move very close to the speaker. In other words, they rarely moved if they had not identified that the signal was worthy of attention.

We have changed the sentence to “The probability of the females approaching the loudspeaker increases as the separation angle between the target and the noise loudspeakers increases”.

Figure 4 – I’m confused about the 1&2 on the timeline. Do you alternate between the two loudspeakers? If so, maybe put ‘1’ and ‘2’ next to the two blue speakers for clarity – and also mention them in the description of 4C. Alternately, if you are mentioning two playbacks from the same loudspeaker, please state that. Either way, did you always do left then right (or vice versa) or did the conditions differ across animals? Did you always do separated first, then coincident? Or did you randomize across subjects? Please state in methods.

The 1 and 2 on the timeline in Figure 4 indicate the possibility of playing the same signal a second time, in case the crocodile has not moved 90 seconds after the end of the first transmission. For the sake of clarity, we have removed this indication from the timeline. As stated in the methods, we also added a sentence in the figure legend: “The same signal could be played again from the same loudspeaker if the crocodile had not moved 90 seconds after the end of the first transmission”. The same information was added for Figure 7.

Regarding randomization, we randomized all playbacks across animals: the condition (co-located vs. separated) and the SNR at which the target signal was played were randomised. Similarly, the position of the loudspeakers was randomly changed for each animal. We have added this information in the methods, lines 389-390: “The first target signal was broadcast by one of the three randomly selected target loudspeakers (co-located or separated), at a random SNR value.”

Fig 6 caption, line 4 – I’d delete ‘higher’ behavioral response here. It isn’t like experiment 1 where you rank responses. Here they just elicited any response. You can say probability of detection increases with SNR, which is accurate. Also in this figure – do the dots and squares along the top and bottom represent individual trials? Subjects?

We fully agree with the referee. The sentence has been modified as suggested: “The probability of signal detection increases with SNR in both conditions (...)”.

Indeed, the dots and squares represent individual trials, all subjects combined. This is now clarified in the figure legend as follows:

“(…) curves = fitted probabilities of behavioral scores: mean of posterior distribution and 95% credible intervals, green dots represent individual trials in co-located condition, blue squares are individual trials in separated condition)”.

This missing information was also added in the legend of Figure 9.

Line 179, change ‘for’ to ‘in’

Done.

Figure 9 caption – again put an apostrophe at the end of loudspeakers or delete the ‘s’. Again, please describe dots and squares along the top and bottom of the figure.

Done.

Line 287 – was the white noise a looping frozen sample? A non-repeating stream? Please specify.

In all experiments, the white noise was a two-hour sequence played in a loop. We have added this information in the methods:

- Lines 308-310: “The masking noise was a white noise (2 hours duration, frequency range [20, 20000] Hz; 83 unweighted dB SPL measured at 1 m using a Sound level meter AMPROBE SM-10; slow time window equal to 1 second). It was broadcast in a loop for the duration of each experimental session.”

- Lines 365-366: “The masking noise (white noise, duration 2 hours) was played continuously in a loop, starting before putting the crocodile in the pond and throughout each experimental session”

-Lines 426-427: “The masking noise (white noise, duration 2 hours) was played continuously in a loop before the tested subject was placed in the experimental room and throughout each trial.”

Line 298 – what were the frequency characteristics of the speakers?

We have added a Supplementary Figure 6 with technical specifications of the speakers, as follows:

A Frequency response of the FoxPro Fusion speakers (rear loudspeaker) used in Experiments 1 and 2.

B Directivity pattern of the FoxPro Fusion speakers (rear loudspeaker) used in Experiments 1 and 2.

C Frequency response of the AudioPro, Bravo Allroom Sat speakers used in Experiment 3.
 D Directivity pattern of the FoxPro Fusion speakers (rear loudspeaker) used in Experiments 4.
 All measurements were performed in a semi-anechoic chamber with calibrated chains.

Line 310 – change ‘mask’ to ‘masker’ (and in other instances where you wrote mask), add a ‘to’ in after ‘raised’

Done.

Line 430 – how many sessions did you have to exclude for this reason?

We had to exclude only one session in which the crocodile did not react to any stimulus, even with a high SNR (no reaction to a series of stimuli at SNR = [-18, -16, -20, -16, -18] dB). We added this information in the methods, line 480: “If the crocodile did not respond more than twice to one of the higher SNRs (-16 dB or -18 dB) in the same session, the entire session was excluded from the final data set, considering that the motivation to respond to the target signal was not sufficient (only one session had to be excluded).”

Reviewer #1 (Remarks to the Author):

The authors have adequately responded to my previous comments.